# On Ulam Stability and Multiplicity Results to a Nonlinear Coupled System with Integral Boundary Conditions

**Kamal Shah [1], Poom Kumam [2],* and Inam Ullah [1]**

[1] Department of Mathematics, University of Malakand, Chakdara Dir(L), Khyber Pakhtunkhwa 18800, Pakistan; kamalshah408@gmail.com (K.S.); inamullahkhan829@gmail.com (I.U.)

[2] KMUTT Fixed Point Research Laboratory, KMUTT-Fixed Point Theory and Applications Research Group, Theoretical and Computational Science Center (TaCS), Science Laboratory Building, Faculty of Science, King Mongkut's University of Technology Thonburi (KMUTT), 126 Pracha-Uthit Road, Bang Mod, Thrung Khru, Bangkok 10140, Thailand

\* Correspondence: poom.kum@kmutt.ac.th or poom.kumam@mail.kmutt.ac.th

**Abstract:** This manuscript is devoted to establishing existence theory of solutions to a nonlinear coupled system of fractional order differential equations (FODEs) under integral boundary conditions (IBCs). For uniqueness and existence we use the Perov-type fixed point theorem. Further, to investigate multiplicity results of the concerned problem, we utilize Krasnoselskii's fixed-point theorems of cone type and its various forms. Stability analysis is an important aspect of existence theory as well as required during numerical simulations and optimization of FODEs. Therefore by using techniques of functional analysis, we establish conditions for Hyers-Ulam (HU) stability results for the solution of the proposed problem. The whole analysis is justified by providing suitable examples to illustrate our established results.

**Keywords:** arbitrary order differential equations; multiple positive solution; Perov-type fixed point theorem; HU stability

## 1. Introduction

Fractional order differential equations (FODEs) emerge in the scientific demonstration of numerous frameworks and different fields of science such as physics, chemistry , economics, polymer rheology, aerodynamics, electrodynamics of complicated medium, blood flow phenomena, biophysics, etc. (see [1–5]). Recently, many authors have studied FODEs from different aspects, one is the numerical and scientific techniques for finding solutions and the other is the theoretical perspective of uniqueness and existence of solutions. The interest of the researchers in the investigation of FODEs lies in the incontrovertible fact that fractional-order models (FOM) are found to be highly realistic and practical, compared to the integer order models. Because there are additional degrees of opportunity in the FOM, in consequence, the subject of FODEs is gaining more attention from researchers. Another facet of research, which has been completely studied for integer order differential equations is devoted to uniqueness and existence of solutions to boundary value problems (BVPs). The mentioned aspect has been very well studied for FODEs, we refer the readers [6–10]. Uniqueness and existence results of solutions to multi-point BVPs have been studied via classical fixed point theorems such as the Schauder fixed point theorem and the Banach contraction principle, see [11–17].

FODEs under integral boundary conditions (IBCs) have been investigated very well because these type of equations are increasingly used in fluid-mechanics and dynamical problems. Jankowski [18] studied the ordinary differential equation under IBCs given by

$$\begin{cases} y'(\vartheta) = F(\vartheta, y(\vartheta)), & \vartheta \in [0, T], \quad T > 0, \\ y(\vartheta)|_{\vartheta=0} = \delta \int_0^T y(s)ds + d_0, & d_0 \in R, \end{cases}$$

where $F \in C([0, T] \times R, R)$ and $\delta = 1$ or $-1$. He developed a sufficient condition for iterative approximate solutions to the above problem.

Nanware and Dhaigude [19] have investigated the aforementioned BVP under the IBCs for FODE as given by

$$\begin{cases} D_{+0}^\sigma y(\vartheta) = F(\vartheta, y(\vartheta)), & \vartheta \in [0, T], \quad T > 0, \\ y(\vartheta)|_{\vartheta=0} = \delta \int_0^T y(s)ds + d_0, & d_0 \in R, \end{cases}$$

where $0 < \sigma \leq 1, \delta$ is 1 or $-1$ and $F \in C([0, T] \times R, R)$, $D_{+0}^\sigma$ is Riemann-Liouville fractional derivative of order $\sigma$ is defined in (2). The aforementioned author also studied the iterative approximate solution to the above FODEs.

In the same line Cabada and Wang [20] studied the following problem under IBCs as

$$\begin{cases} {}^C D_{+0}^\sigma y(\vartheta) + \varphi(\vartheta, y(\vartheta)) = 0; & \vartheta \in (0, 1), \\ y(0) = y''(0) = 0, & y(1) = \delta \int_0^1 y(s)ds, \end{cases}$$

where $\sigma \in (2, 3], \delta \in (0, 2)$ and $y : [0, 1] \times [0, \infty] \to [0, \infty]$ are the continuous functions. Also we remark that ${}^C D_{+0}^\sigma$ stands for Caputo's fractional derivative.

Inspired from the aforementioned work, in this article we investigate a system of nonlinear FODEs with IBCs as

$$\begin{cases} D_{+0}^\sigma y(\vartheta) + \varphi(\vartheta, y(\vartheta), z(\vartheta)) = 0; & \vartheta \in (0, 1); \quad m - 1 < \sigma \leq m, \\ D_{+0}^{æ} z(\vartheta) + \chi(\vartheta, y(\vartheta), z(\vartheta)) = 0; & \vartheta \in (0, 1); \quad m - 1 < æ \leq m, \\ y(0) = y'(0) = y''(0) = \cdots = y^{(m-2)}(0) = 0, & y(1) = \delta \int_0^1 y(s)ds, \\ z(0) = z'(0) = z''(0) = 0 \cdots = z^{(m-2)}(0) = 0, & z(1) = \varrho \int_0^1 z(s)ds, \end{cases} \tag{1}$$

such that $m \geq 3, \delta, \varrho \in (0, 2)$, the functions $\varphi, \chi : [0, 1] \times [0, \infty] \times [0, \infty] \to [0, \infty]$ are continuous functions and $D_{+0}^\sigma, D_{+0}^{æ}$ stand for Riemann-Liouville fractional derivatives is defined in (2). We claim that such a system of FODEs are very rarely considered for stability as well as multiplicity results. Our analysis is devoted to the existence theory of a solution, multiplicity results and stability analysis of the suggested problem.

During the last few decades another part of research, which has been considered for FODEs and got much attention from the researchers is stability analysis. Numerous forms of stabilities have been studied in literature which are Mittag-Leffer stability, exponential stability, Lyapunov stability etc., we refer [21–23].

The Ulam stability was first presented by Ulam in 1940 and then brilliantly explained by Hyers in 1941. For more information about HU stability, we refer [24,25]. The HU stability results were generalized and extended by many researchers for FODEs under IBCs. In 1978, Jung studied the said stability for ODEs. Oblaz, Benchohra, etc., have studied the said stability for FODEs but their investigation was limited to initial value problems, we refer to [26–28]. To the best of our information and knowledge, the HU stability has been very rarely studied for coupled system of FODEs under

IBCs. Therefore in this article we investigate HU stability to the considered problem. Here we remark that we also provide some necessary results for nonexistence of solution. Finally a series of examples are provided to support our analysis.

## 2. Axillary Results

In the current section, we review some fundamental definitions and useful results of functional analysis, fractional calculus and fixed point theory (see reference [1,2,8,29–32]). Here, first of all, we define the Banach space which is utilized throughout in this article.

Let us define $E = \{y(\vartheta)|y \in C[0,1]\}$ with the norm $||y|| = \max_{\vartheta \in [0,1]} |y(\vartheta)|$. We define the norm for the product space as $||(y,z)|| = ||y|| + ||z||$. Obviously $(E \times E, || \cdot ||)$ is a Banach space. Let $K = [\theta, 1 - \theta]$ for each $\theta \in (0,1)$, then, we define the cone $C \subset E \times E$ by

$$C = \{(y,z) \in E \times E : \min_{\vartheta \in K}[y(\vartheta) + z(\vartheta)] \geq \lambda ||(y,z)||\}.$$

$$C_r = \{(y,z) \in C : ||(y,z)|| \leq r\}, \partial C_r = \{(y,z) \in K : ||(y,z)|| = r\}.$$

As in [31], we define positive solution as follows.

**Definition 1.** *A pair of functions* $(y,z) \in E \times E$ *is called a positive solution of problem* (1) *under the given IBCs if* $D^{\sigma}_{+0}y, D^{\rho}_{+0}z \in L^1[0,1]$ *with* $(y,z) > (0,0)$ *on* $(0,1] \times (0,1]$, *where the functions* $y,z$ *satisfy the IBCs given in* (1) *respectively, for all* $\vartheta \in [0,1]$.

**Definition 2.** *The Riemann-Liouville fractional derivative of order* $\sigma > 0$ *of a continuous function* $y : (0,\infty) \to R$ *is defined as*

$$D^{\sigma}_{+0}y(\vartheta) = \frac{1}{\Gamma(m-\sigma)}\left(\frac{d}{d\vartheta}\right)^m \int_0^{\vartheta}(\vartheta - s)^{m-\sigma-1}y(s)ds, \tag{2}$$

*where* $m = [\sigma] + 1$ *and* $[\sigma]$ *denotes the integer part of* $\sigma$.

**Definition 3.** *The Riemann-Liouville fractions of integration of order* $\sigma > 0$ *of a continuous function* $y : (0,\infty) \to R$ *is defined by*

$$I^{\sigma}_{+0}y(\vartheta) = \frac{1}{\Gamma(\sigma)}\int_0^{\vartheta}(\vartheta - s)^{\sigma-1}y(s)ds, \tag{3}$$

*where the integral is point-wise defined on* $(0,\infty)$.

**Lemma 1.** *Let* $\sigma > o$, *then the FODE*

$$D^{\sigma}_{+0}y(\vartheta) = 0 \tag{4}$$

*has a solution given by*

$$y(\vartheta) = \sum_{i=1}^{m} \frac{y^i(0)}{i!}\vartheta^{-i}. \tag{5}$$

**Lemma 2.** *Let* $\sigma > 0$. *Then we have*

$$I^{\sigma}_{+0}[D^{\sigma}_{+0}y(\vartheta)] = y(\vartheta) - \sum_{i=0}^{m} \frac{y^i(0)}{i!}\vartheta^{-i}. \tag{6}$$

**Lemma 3.** *[2] Let* $\sigma > o$ *and* $\vartheta \in C(0,1) \cap L(0,1)$, *then the FODE*

$$D^{\sigma}_{+0}y(\vartheta) = h(\vartheta)$$

*has a solution given by*

$$y(\vartheta) = c_1 \vartheta^{\sigma-1} + c_2 \vartheta^{\sigma-2} + \cdots + c_m \vartheta^{\sigma-m} + I_{+0}^{\sigma} h(\vartheta),$$

*where* $c_i \in R$ *for* $i = 0, 1, 2, \ldots, m$ *and* $m = [\sigma] + 1$.

**Definition 4.** *[32,33] On the Banach space* E *defined afore, the mapping* $d : E \times E \to R^n$ *is called a generalized metric on* E *if* $\forall$ x, y, *and* $y, z \in E$ *with* $y \neq x, z \neq y, z \neq y$, *then the following hold*

$(A1)$ $d(y, z) = 0 \Leftrightarrow y = z, \ \forall \ y, z \in E$
$(A2)$ $d(y, z) = d(z, y), \ \forall \ y, z \in E$
$(A3)$ $d(x, y) = d(x, z) + d(z, y) + d(y, y), \forall \ x, y, y, z, \in E.$

*Further the pair* $(E, d)$ *is called a generalized metric space.*

**Definition 5.** *[32,33] Let* $M = \{M_{m,m} \in R_+^{m \times m}\}$, *for any matrix* $\mathbf{B} \in M$ *the spectral radius is defined by* $æ(\mathbf{B}) = \sup\{|\widehat{\lambda}_i|, i = 1, 2, ..., m\}$, *where* $\widehat{\lambda}_i$, *for* $i = 1, 2, ..., m$ *are the eigenvalues of the matrix* $\mathbf{B}$ *and the matrix will converge to zero if* $æ(\mathbf{B}) < 1$.

**Lemma 4.** *[32,33] A complete generalized metric space* $(M, d)$, *with operator* $B : M \to M$ *such that there* $\exists$ *a matrix* $\mathbf{B} \in M$ *with*

$$d(By, Bz) \leq Bd(y, z), \text{for all } y, z \in M,$$

*if* $æ(\mathbf{B}) < 1$, *then* B *has a fixed point in* M.

**Lemma 5.** *[32,33] Consider a Banach space* E *with cone* $C \subseteq E$ *and* $y \subset C$ *is relatively open set with* $0 \in y$ *and* $B : \bar{y} \to y$ *be a completely continuous mapping. Then one of the following hold*

$(A1)$ *The mapping* B *has a fixed point in* $\bar{y}$
$(A2)$ *There exist* $y \in \partial y$ *and* $\eta \in (0, 1)$ *with* $y = \eta By$.

**Lemma 6.** *[33,34] Consider a cone* C *in the Banach space* E *and if* $\mathfrak{A}_1$ *and* $\mathfrak{A}_2$ *be two bounded open sets in* E, *such that* $0 \in \mathfrak{A}_1 \subset \overline{\mathfrak{A}}_1 \subset \mathfrak{A}_2$. *Let* $B : C \cap (\overline{\mathfrak{A}}_2 \smallsetminus \mathfrak{A}_1) \to C$ *be completely continuous operator and one of the following satisfied:*

(1)    $\|By\| \leq \|y\| \ \forall \ y \in C \cap \partial \mathfrak{A}_1; \|B\| \geqslant \|y\|, \forall \ y \in C \cap \partial \mathfrak{A}_2$
(2)    $\|By\| \geqslant \|y\| \ \forall \ y \in C \cap \partial \mathfrak{A}_1; \|B\| \leq \|y\|, \forall \ y \in C \cap \partial \mathfrak{A}_2$

*Then* B *has at least one fixed point in* $C \cap (\overline{\mathfrak{A}}_2 \smallsetminus \mathfrak{A}_1)$.

## 3. Existence of at Least One Solution

**Lemma 7.** *Let* $h \in C[0, 1]$, *then the BVP*

$$\begin{cases} D_{+0}^{\sigma} y(\vartheta) + h(\vartheta) = 0; & \vartheta \in (0, 1); \quad m - 1 < \sigma \leq m, \\ y(0) = y'(0) = y''(0) = \cdots = y^{(m-2)}(0) = 0, & y(1) = \delta \int_0^1 y(s) ds, \end{cases} \tag{7}$$

*where* $\delta \in (0, 2)$, *has the following unique solution*

$$y(\vartheta) = \int_0^1 H_{\sigma}(\vartheta, s) h(s) ds,$$

where $H_\sigma$ is the Green's function given by

$$H_\sigma(\vartheta, s) = \begin{cases} \dfrac{\vartheta^{\sigma-1}(1-s)^{\sigma-1}(\sigma-\delta+\delta s) - (\sigma-\delta)(\vartheta-s)^{\sigma-1}}{(\sigma-\delta)\Gamma(\delta)}, & 0 \le s \le \vartheta \le 1, \\[4mm] \dfrac{\vartheta^{\sigma-1}(1-s)^{\sigma-1}(\sigma-\delta+\delta s)}{(\sigma-\delta)\Gamma(\delta)}, & 0 \le \vartheta \le s \le 1. \end{cases} \tag{8}$$

**Proof.** Thanks to Lemma 3 for (7), one has

$$y(\vartheta) = -I_{+0}^\sigma h(\vartheta) + c_1 \vartheta^{\sigma-1} + c_2 \vartheta^{\sigma-2} + \cdots + c_m \vartheta^{\sigma-m}. \tag{9}$$

By using initial condition $y(0) = y'(0) = y''(0) = \cdots = y^{(m-2)}(0) = 0$, we get $c_2 = c_3 = \cdots = c_m = 0$. Therefore (9) implies that

$$y(\vartheta) = c_1 \vartheta^{\sigma-1} - I_{+0}^\sigma h(\vartheta). \tag{10}$$

By using boundary condition $y(1) = \delta \int_0^1 y(s)ds$ in (10), we get

$$c_1 = \int_0^1 \frac{(\vartheta-s)^{\sigma-1}}{\Gamma(\sigma)} h(s)ds + \delta \int_0^1 y(s)ds.$$

Hence we have the following solution to (1)

$$y(\vartheta) = -\int_0^\vartheta \frac{(\vartheta-s)^{\sigma-1}}{\Gamma(\sigma)} h(s)ds + \vartheta^{\sigma-1} \int_0^1 \frac{(1-s)^{\sigma-1}}{\Gamma(\sigma)} h(s)ds + \delta\vartheta^{\sigma-1} \int_0^1 y(s)ds. \tag{11}$$

Let $B = \int_0^1 y(s)ds$, then from Equation (11), we have

$$B = -\int_0^1 \int_0^\vartheta \frac{(\vartheta-s)^{\sigma-1}}{\Gamma(\sigma)} h(s)ds + \int_0^1 \int_0^1 \frac{\vartheta^{\sigma-1}(\vartheta-s)^{\sigma-1}}{\Gamma(\sigma)} h(s)ds + \int_0^1 \delta B \vartheta^{\sigma-1} ds$$

$$B = -\int_0^1 \frac{(1-s)^\sigma}{\sigma\Gamma(\sigma)} h(s)ds + \int_0^1 \frac{(1-s)^{\sigma-1}}{\sigma\Gamma(\sigma)} h(s)ds + \frac{1}{\sigma}\delta B \tag{12}$$

implies Equation (12), so we get

$$B = -\frac{1}{\sigma-\delta} \int_0^1 \frac{(1-s)^\sigma}{\Gamma(\sigma)} h(s)ds + \frac{1}{\sigma-\delta} \int_0^1 \frac{(1-s)^{\sigma-1}}{\Gamma(\sigma)} h(s)ds.$$

Replacing this valve in (11), we get

$$y(\vartheta) = -\int_0^t \frac{(\vartheta-s)^{\sigma-1}}{\Gamma(\sigma)} h(s)ds + \vartheta^{\sigma-1} \int_0^1 \frac{(1-s)^{\sigma-1}}{\Gamma(\sigma)} h(s)ds - \frac{\delta}{\sigma-\delta} \int_0^1 \frac{\vartheta^{\sigma-1}(1-s)^\sigma}{\sigma\Gamma(\sigma)} h(s)ds$$

$$+ \frac{\delta}{\sigma-\delta} \int_0^1 \frac{\vartheta^{\sigma-1}(1-s)^{\sigma-1}}{\Gamma(\sigma)} h(s)ds.$$

$$= -\int_0^\vartheta \frac{(\vartheta-s)^{\sigma-1}}{\Gamma(\sigma)} h(s)ds + \int_0^1 \frac{\vartheta^{\sigma-1}(1-s)^{\sigma-1}(\sigma-\delta+\delta s)}{(\sigma-\delta)\Gamma(\sigma)} h(s)ds$$

$$= \int_0^\vartheta \frac{\vartheta^{\sigma-1}(1-s)^{\sigma-1}(\sigma-\delta+\delta s) - (\sigma-\delta)(\vartheta-s)^{\sigma-1}}{(\sigma-\delta)\Gamma(\sigma)} h(s)ds$$

$$+ \int_\vartheta^1 \frac{\vartheta^{\sigma-1}(1-s)^{\sigma-1}(\sigma-\delta+\delta s)}{(\sigma-\delta)\Gamma(\sigma)} h(s)ds$$

$$= \int_0^1 H_\sigma(\vartheta, s) h(s)ds,$$

where $H_\sigma(\vartheta, s)$ is the Green's function of BVP (7). Similarly we can obtain $z(\vartheta) = \int_0^1 H_{\text{æ}}(\vartheta, s)h(s)ds$, where $H_{\text{æ}}(\vartheta, s)$ is the Green's function for the second equation of the system (1) and is given by

$$H_{\text{æ}}(\vartheta, s) = \begin{cases} \dfrac{\vartheta^{\text{æ}-1}(1-s)^{\text{æ}-1}(\text{æ} - \varrho + \varrho s) - (\text{æ} - \varrho)(\vartheta - s)^{\text{æ}-1}}{(\text{æ} - \varrho)\Gamma(\text{æ})}, & 0 \le s \le \vartheta \le 1, \\[3mm] \dfrac{\vartheta^{\text{æ}-1}(1-s)^{\text{æ}-1}(\text{æ} - \varrho + \varrho s)}{(\text{æ} - \varrho)\Gamma(\text{æ})}, & 0 \le \vartheta \le s \le 1. \end{cases} \tag{13}$$

$\square$

**Lemma 8.** *Let* $H(\vartheta, s) = (H_\sigma(\vartheta, s), H_{\text{æ}}(\vartheta, s))$ *be the Green's function of* (1) *defined in Equations* (8) *and* (13). *This* $H(\vartheta, s)$ *has the given properties*

$(F_1)$ $H(\vartheta, s)$ *is continuous function on the unit square* $\forall (\vartheta, s) \in [0, 1] \times [0, 1]$
$(F_2)$ $H(\vartheta, s) \ge 0 \ \forall \vartheta, s \in [0, 1]$ *and* $H(\vartheta, s) > 0 \ \forall \vartheta, s \in (0, 1)$
$(F_3)$ $\max\limits_{0 \le \vartheta \le 1} H(\vartheta, s) = H(1, s), \forall s \in [0, 1]$
$(F_4)$ $\min\limits_{\vartheta \in [\theta, 1-\theta]} H(\vartheta, s) \ge \lambda(s)H(1, s)$ *for each* $\theta$, $s \in (0, 1)$,

*where* $\lambda = \min\{\lambda_\sigma = \theta^{\sigma-1}, \lambda_{\text{æ}} = \theta^{\text{æ}-1}\}$.

Now according to Lemma 7, we can write system (1) as follows

$$\begin{cases} y(\vartheta) = \displaystyle\int_0^1 H_\sigma(\vartheta, s)\varphi(s, y(s), z(s))ds, \\[3mm] z(\vartheta) = \displaystyle\int_0^1 H_{\text{æ}}(\vartheta, s)\chi(s, y(s), z(s))ds. \end{cases} \tag{14}$$

Let $B : E \times E \to E \times E$ be the operator defined as

$$\begin{aligned} B(y, z)(\vartheta) &= \left( \int_0^1 H_\sigma(\vartheta, s)\varphi(s, y(s), z(s))ds, \ \int_0^1 H_{\text{æ}}(\vartheta, s)\chi(s, y(s), z(s))ds \right). \\ &= \left( B_1(y, z)(\vartheta), B_2(y, z)(\vartheta) \right). \end{aligned} \tag{15}$$

Then the fixed point of operator $B$ coincides with the solution of the coupled system (1).

**Theorem 1.** *Consider that* $u, v : [0, 1] \times [0, \infty) \times [0, \infty) \to [0, \infty)$ *are continuous. Then* $B(C) \subset C$ *and* $B : C \to C$ *is completely continuous, where* $B$ *is defined in* (15).

**Proof.** To prove that $B(C) \subset C$, let $(y, z) \in C$, then by Lemma 8, we have $B(y, z) \in C$ and from $(F_4)$ and $\forall \vartheta \in K$, we obtain

$$B_1(y(\vartheta), z(\vartheta)) = \int_0^1 H_\sigma(\vartheta, s)\varphi(s, y(s), z(s))ds \ge \lambda_\sigma \int_0^1 H_\sigma(1, s)\varphi(s, y(s)), z(s)ds. \tag{16}$$

Also from $(F_3)$, we obtain

$$B_1(y(\vartheta), z(\vartheta)) = \int_0^1 H_\sigma(\vartheta, s)\varphi(s, y(s), z(s))ds \le \int_0^1 H_\sigma(1, s)\varphi(s, y(s)), z(s)ds. \tag{17}$$

Thus from (16) and (17), we have

$$B_1(y(\vartheta), z(\vartheta)) \geq \lambda \|B_1(y, z)\|, \text{ for all } \vartheta \in K.$$

Similarly, one can write that

$$B_2(y(\vartheta), z(\vartheta)) \geq \lambda \|B_2(y, z)\|, \text{ for all } \vartheta \in K.$$

Thus

$$B_1(y(\vartheta), z(\vartheta)) + B_2(y(\vartheta), z(\vartheta)) \geq \lambda \|B(y, z)\|, \text{ for all } \vartheta \in K,$$

$$\min_{\vartheta \in K}[B_1(y(\vartheta), z(\vartheta)) + B_2(y(\vartheta), z(\vartheta))] \geq \lambda \|B(y, z)\|.$$

Hence we have $B(y, z) \in C \Rightarrow B(C) \subset C$. Next, like the proof of Theorem 1 of [35], and applying the Arzelà-Ascoli's theorem, it can be easily proven that $B : C \to C$ is completely continuous $\square$

**Theorem 2.** *Consider that $\varphi$ and $\chi$ are continuous on $[0,1] \times [0, \infty) \times (0, \infty) \to [0, \infty)$, and there exist $f_i(\vartheta), H_i(\vartheta), (i = 1, 2) : (0, 1) \to [0, \infty)$ that satisfy*

$(A_1)$ $|\varphi(\vartheta, y, z) - \varphi(\vartheta, \bar{y}, \bar{z})| \leq u_1(\vartheta)|y - \bar{y}| + v_1(\vartheta)|z - \bar{z}|$, *for $\vartheta \in (0, 1)$ and $y, z, \bar{y}, \bar{z} \geq 0$*
$(A_2)$ $|\chi(\vartheta, y, z) - \chi(\vartheta, \bar{y}, \bar{z})| \leq u_2(\vartheta)|y - \bar{y}| + v_2(\vartheta)|z - \bar{z}|$, *for $\vartheta \in (0, 1)$ and $y, z, \bar{y}, \bar{z} \geq 0$*
$(A_3)$ $æ(\mathbf{B}) < 1$, *where $\mathbf{B} \in \{M_{2,2} \in R_+^{2 \times 2}\}$ is a matrix given by*

$$\begin{bmatrix} \int_0^1 H_\sigma(1, s)u_1(s)ds & \int_0^1 H_\sigma(1, s)v_1(s)ds \\ \int_0^1 H_æ(1, s)u_2(s)ds & \int_0^1 H_æ(1, s)v_2(s)ds \end{bmatrix}.$$

*Then the system (1) has a unique positive solution $(y, z) \in C$.*

**Proof.** Let us define a generalized metric $d : E^2 \times E^2 \to R^2$ by

$$d((y, z), (\bar{y}, \bar{z})) = \begin{pmatrix} \|y - \bar{y}\| \\ \|z - \bar{z}\| \end{pmatrix}, \text{ for all } (y, z), (\bar{y}, \bar{z}) \in E \times E.$$

Obviously $(E \times E, d)$ is a generalized complete metric space. Then for any $(y, z), (\bar{y}, \bar{z}) \in E \times E$ and using property $(F_3)$ we get

$$|B_1(y, z)(\vartheta) - B_1(\bar{y}, \bar{z})(\vartheta)| \leq \max_{\vartheta \in [0,1]} \int_0^1 |H_\sigma(\vartheta, s)|[|\varphi(s, y(s), z(s)) - \varphi(s, \bar{y}(s), \bar{z}(s))|]ds$$

$$\leq \int_0^1 H_\sigma(1, s)[u_1(s)\|y - \bar{y}\| + v_1(s)\|z - \bar{z}\|]ds$$

$$\leq \int_0^1 u_1(s)H_\sigma(1, s)ds\|y - \bar{y}\| + \int_0^1 v_1(s)H_\sigma(1, s)ds\|z - \bar{z}\|.$$

Similarly we can show that

$$|B_2(y, z) - B_2(\bar{y}, \bar{z})| \leq \int_0^1 u_2(s)H_æ(1, s)ds\|y - \bar{y}\| + \int_0^1 v_2(s)H_æ(1, s)ds\|z - \bar{z}\|.$$

Thus we have

$$|B(y, z) - B(\bar{y}, \bar{z})| \leq \mathbf{B}d((y, z), (\bar{y}, \bar{z})), \forall(y, z), (\bar{y}, \bar{z}) \in E \times E,$$

where

$$\mathbf{B} = \begin{bmatrix} \int_0^1 H_\sigma(1,s)u_1(s)ds & \int_0^1 H_\sigma(1,s)v_1(s)ds \\ \int_0^1 H_\mathit{æ}(1,s)u_2(s)ds & \int_0^1 H_\mathit{æ}(1,s)v_2(s)ds \end{bmatrix}.$$

As $\mathit{æ}(\mathbf{B}) < 1$, in the light of Lemma 4, system (1) has a unique positive solution. $\square$

**Theorem 3.** *Consider that $\varphi$ and $\chi$ are continuous on $[0,1] \times [0,\infty) \times (0,\infty) \to [0,\infty)$ and there exist $a_i, b_i, c_i (i = 1,2) : (0,1) \to [0,\infty)$ satisfying:*

$(A_4)$ $\varphi(\vartheta, y(\vartheta), z(\vartheta)) \le a_1(\vartheta) + b_1(\vartheta)y(\vartheta) + c_1(\vartheta)z(\vartheta)$, $\vartheta \in (0,1)$, $y, z \ge 0$

$(A_5)$ $\chi(\vartheta, y(\vartheta), z(\vartheta)) \le a_2(\vartheta) + b_2(\vartheta)y(\vartheta) + c_2(\vartheta)z(\vartheta)$, $\vartheta \in (0,1)$, $y, z \ge 0$

$(A_6)$ $\Lambda_1 = \int_0^1 H_\sigma(1,s)a_1(s)ds < \infty$, $\Delta_1 = \int_0^1 H_\sigma(1,s)[b_1(s) + c_1(s)]ds < \frac{1}{2}$

$(A_7)$ $\Lambda_2 = \int_0^1 H_\mathit{æ}(1,s)a_2(s)ds < \infty$, $\Delta_2 = \int_0^1 H_\mathit{æ}(s,s)[b_2(s) + c_2(s)]ds < \frac{1}{2}$.

*Then the system (1) has at least one positive solution in*

$$\left\{ (y,z) \in C : \|(y,z)\| \le r \right\}, \text{ where } \max\left\{ \frac{\Lambda_1}{\frac{1}{2} - \Delta_1}, \frac{\Lambda_2}{\frac{1}{2} - \Delta_2} \right\} < r.$$

**Proof.** Define $\Omega = \left\{ (y,z) \in C : \|(y,z)\| < r \right\}$ with $\max\left\{ \frac{\Lambda_1}{\frac{1}{2} - \Delta_1}, \frac{\Lambda_2}{\frac{1}{2} - \Delta_2} \right\} < r.$

According to the Theorem 1, the operator $B : \overline{\Omega} \to C$ is completely continuous. Let $(y,z) \in \Omega$, such that $\|(y,z)\| < r$. Then, we have

$$\|B_1(y,z)\| = \max_{\vartheta \in [0,1]} \left| \int_0^1 H_\sigma(\vartheta, s)\varphi(s, y(s), z(s)) \right| ds$$

$$\le \left( \int_0^1 H_\sigma(1,s)a_1(s)ds + \int_0^1 H_\sigma(1,s)b_1(s)|y(s)|ds + \int_0^1 H_\sigma(1,s)c_1(s)|z(s)|ds \right)$$

$$\le \int_0^1 H_\sigma(1,s)a_1(s)ds + r\left[ \int_0^1 H_\sigma(1,s)[b_1(s) + c_1(s)]ds \right]$$

$$\le \Lambda_1 + r\Delta_1 < \frac{r}{2},$$

Similarly, $\|B_2(y,z)\| < \frac{r}{2}$, thus $\|B(y,z)\| < r$. Therefore, thanks to Lemma 5, we have $B(y,z) \in \overline{\Omega}$, thus $B : \overline{\Omega} \to \overline{\Omega}$. Let there exist $\varsigma \in (0,1)$ and $(y,z) \in \partial\Omega$ such that $(y,z) = \varsigma B(y,z)$. Then in the light of assumptions $(A_4), (A_5)$ and by $(F_4)$ of Lemma 8, we get $\forall \vartheta \in [0,1]$

$$|z(\vartheta)| \le \varsigma \int_0^1 H_\sigma(\vartheta, s)|\varphi(s, y(s), z(s))|ds$$

$$\le \varsigma \left( \int_0^1 H_\sigma(1,s)a_1(s)ds + \int_0^1 H_\sigma(1,s)b_1(s)y(s)ds + \int_0^1 H_\sigma(1,s)c_1(s)z(s)ds \right)$$

$$\le \varsigma \left( \Delta_1 + r\Lambda_1 \right)$$

$$< \varsigma \frac{r}{2}$$

which implies that $\|y\| < \varsigma \frac{r}{2}$. Similarly, it can be proved that $\|z\| < \varsigma \frac{r}{2}$. From which, we have $\|(y,z)\| < \varsigma r$, with $\varsigma \in (0,1)$ which is a contradiction that $(y,z) \in \partial\Omega$ as $r = \|(y,z)\|$. Thus, according to Lemma 5, B has at least one fixed point $(y,z) \in \overline{\Omega}$. $\square$

Next the following assumptions and notations will be used:

$(C_1)$ $\varphi, \chi : [0,1] \times [0,\infty) \times [0,\infty) \to [0,\infty)$ are continuous and $\varphi(\vartheta,0,0) = \chi(\vartheta,0,0) = 0$ uniformly with respect to $\vartheta$ on $[0,1]$

$(C_2)$ $H_\sigma(1,s), H_{\ae}(1,s)$ defined in Lemma 8 satisfy

$$0 < \int_0^1 H_\sigma(1,s)ds < \infty, \ 0 < \int_0^1 H_{\ae}(1,s)ds < \infty$$

$(C_3)$ Let these limits hold

$$
\begin{aligned}
\varphi^\alpha &= \lim_{(y,z)\to(\alpha,\alpha)} \sup_{\vartheta\in[0,1]} \frac{\varphi(\vartheta,y,z)}{y+z}, \ \chi^\alpha = \lim_{(y,z)\to(\alpha,\alpha)} \sup_{\vartheta\in[0,1]} \frac{\chi(\vartheta,y,z)}{y+z}, \\
\varphi_\alpha &= \lim_{(y,z)\to(\alpha,\alpha)} \in f_{\vartheta\in[0,1]} \frac{\varphi(\vartheta,y,z)}{y+z}, \ \chi_{\ae} = \lim_{(y,z)\to(\alpha,\alpha)} \in f_{\vartheta\in[0,1]} \frac{\chi(\vartheta,y,z)}{y+z}, \text{ where } \alpha \in \{0,\infty\}.
\end{aligned}
\tag{18}
$$

$$\alpha_\sigma = \max_{\vartheta\in[0,1]} \int_0^1 H_\sigma(\vartheta,s)ds, \ \alpha_{\ae} = \max_{\vartheta\in[0,1]} \int_0^1 H_{\ae}(\vartheta,s)ds. \tag{19}$$

**Theorem 4.** *If the assumptions $(C_1) - (C_2)$ hold and one of the following conditions is also satisfied:*

$(D_1)$ $\varphi_0 \left( \lambda^2 \int_\theta^{1-\theta} H_\sigma(1,s)ds \right) > 1$, $\varphi^\infty \alpha_\sigma < 1$ and $\chi_0 \left( \lambda^2 \int_\theta^{1-\theta} H_{\ae}(1,s)ds \right) > 1$, $\chi^\infty \alpha_{\ae} < 1$.
*Moreover, $\varphi_0 = \chi_0 = \infty$ and $\varphi^\infty = \chi^\infty = 0$*

$(D_2)$ *There exist two constants $\eta_1, \eta_2$ with $0 < \eta_1 \leq \eta_2$ such that $\varphi(\vartheta,\cdot,\cdot)$ and $\chi(\vartheta,\cdot,\cdot)$ are nondecreasing on $[0,\eta_2]$ $\forall \vartheta \in [0,1]$,*

$$\varphi(\vartheta, \lambda_\sigma \eta_1, \lambda_{\ae} \eta_1) \geq \frac{\eta_1}{2} \left( \lambda_\sigma \int_\theta^{1-\theta} H_\sigma(1,s)ds \right)^{-1},$$

$$\chi(\vartheta, \lambda_\sigma \eta_1, \lambda_{\ae} \eta_1) \geq \frac{\eta_1}{2} \left( \lambda_{\ae} \int_\theta^{1-\theta} H_{\ae}(1,s)ds \right)^{-1}$$

*and $\varphi(\vartheta, \eta_2, \eta_2) \leq \frac{\eta_2}{2\alpha_\sigma}$, $\chi(\vartheta, \eta_2, \eta_2) \leq \frac{\eta_2}{2\alpha_{\ae}}$, for all $\vartheta \in [0,1]$,*

*where $\lambda, H_\sigma(1,s), H_{\ae}(1,s)$ defined in Lemma 8 and $\varphi_0, \chi_0, \varphi^\infty, \chi^\infty, \alpha_\sigma, \sigma_\alpha$ defined in Equations (18) and (19). Then the coupled system (1) has at least one positive solution.*

**Proof.** B as defined in (15) is completely continuous.

**Case I.** Let the condition $(D_1)$ hold. Taking $\varphi_0 \left( \lambda_\sigma^2 \int_\theta^{1-\theta} H_\sigma(1,s)ds \right) > 1$, then there exists a constant $\kappa_1 > 0$ such that

$$\varphi(\vartheta,y,z) \geq (\varphi_0 - r_1)(y(\vartheta) + z(\vartheta)), \ \chi(\vartheta,y,z) \geq (\chi_0 - r_2)(y(\vartheta) + z(\vartheta)), \text{ for all } \vartheta \in [0,1], y,z \in [0,\kappa_1],$$

where $r_1 > 0$, and satisfies the conditions

$$(\varphi_0 - r_1)\frac{\lambda_\sigma^2}{2}\int\limits_\theta^{1-\theta} H_\sigma(1,s)ds \geq 1, \quad (\chi_0 - r_1)\frac{\lambda_æ^2}{2}\int\limits_\theta^{1-\theta} H_æ(1,s)ds \geq 1.$$

So for $\vartheta \in [0,1], (y,z) \in \partial C_{\kappa_1}$, we have

$$B_1(y,z)(\vartheta) = \int\limits_0^1 H_\sigma(\vartheta,s)\varphi(s,y(s),z(s))ds \geq \lambda_\sigma \int\limits_0^1 H_\sigma(1,s)\varphi(s,y(s),z(s))ds$$

$$\geq (\varphi_0 - r_1)\frac{\lambda_\sigma^2}{2}\int\limits_0^1 H_\sigma(1,s)ds\|(y,z)\| \geq \frac{\|(y,z)\|}{2}.$$

Analogously

$$B_2(y,z)(\vartheta) = \int\limits_0^1 H_æ(\vartheta,s)\chi(s,y(s),z(s))ds \geq \lambda_æ \int\limits_0^1 H_æ(1,s)\varphi(s,y(s),z(s))ds$$

$$\geq (\chi_0 - r_2)\frac{\lambda_æ^2}{2}\int\limits_0^1 H_æ(1,s)ds\|(y,z)\| \geq \frac{\|(y,z)\|}{2}.$$

Therefore, we have

$$\|B(y,z)\| \geq \|B_1(y,z)\| + \|B_2(y,z)\| \geq \|(y,z)\|. \tag{20}$$

Also for $\varphi^\infty \alpha_\sigma < 1$ and $\chi^\infty \alpha_æ < 1$, there exists a constant say $\bar\kappa_2 > 0$ such that $\varphi(\vartheta,y,z) \leq (\varphi^\infty + r_2)(y+z), \chi(\vartheta,y,z) \leq (\chi^\infty + r_2)(y+z)$, for $\vartheta \in [0,1], y, z \in (\bar\kappa_2, \infty)$, where $r_2 > 0$ satisfies the conditions $\alpha_\sigma(\varphi^\infty + r_2) \leq 1, \alpha_æ(\chi^\infty + r_2) \leq 1$. Let $J = \max_{\vartheta \in [0,1], y, z \in [0,\bar\kappa_2]} \varphi(\vartheta,y,z), L = \max_{\vartheta \in [0,1], y, z \in [0,\bar\kappa_2]} \chi(\vartheta,y,z)$, then $\varphi(\vartheta,y,z) \leq J + (\varphi^\infty + r_2)(y,z), \chi(\vartheta,y,z) \leq L + (\chi^\infty + r_2)(y,z)$. Now setting $\max\{\kappa_1, \bar\kappa_2, J\alpha_\sigma(1 - \alpha_\sigma(\varphi^\infty + r_2))^{-1}\} \leq \frac{\kappa_2}{2}, \max\{\kappa_1, \bar\kappa_2, L\alpha_æ(1 - \alpha_æ(\chi^\infty + r_2))^{-1}\} \leq \frac{\kappa_2}{2}$.

So for any $\vartheta \in [0,1], (y,z) \in \partial C_{\kappa_2}$, we obtain

$$B_1(y,z)(\vartheta) = \int\limits_0^1 H_\sigma(\vartheta,s)\varphi(s,y(s),z(s))ds \leq \lambda_\sigma \int\limits_0^1 H_\sigma(1,s)\varphi(s,y(s),z(s))ds$$

$$\leq \int\limits_0^1 H_\sigma(1,s)(J + (\varphi^\infty + r_2)[u(s) + z(s)]ds$$

$$\leq J\int_0^1 H_\sigma(1,s)ds + (\varphi^\infty + r_2)\int\limits_0^1 H_\sigma(1,s)ds\|(y,z)\|$$

$$< \frac{\kappa_2}{2} - \alpha_\sigma(\varphi^\infty + r_2)\frac{\kappa_2}{2} + (\varphi^\infty + r_2)\alpha_\sigma\|(y,z)\| < \frac{\kappa_2}{2}.$$

Similarly $B_2(y,z)(\vartheta) < \frac{\kappa_2}{2}$, as $(y,z) \in \partial C_{\kappa_2}$, thus we have

$$\|B(y,z)\| < \|(y,z)\|. \tag{21}$$

**Case II.** If assumptions in $(D_2)$ hold, then in light of the definition of C for $(y, z) \in \partial C_{\eta_1}$, we have $\|(y, z)\| = \eta_1$, for $\vartheta \in K$. Then from $(D_2)$, we have

$$B_1(y, z)(\vartheta) = \int_0^1 H_\sigma(\vartheta, s) \varphi(s, y(s), z(s)) ds \geq \lambda_\sigma \int_\theta^{1-\theta} H_\sigma(1, s) \varphi(s, y(s), z(s)) ds$$

$$\geq \left( \lambda_\sigma \int_\theta^{1-\theta} H_\sigma(1, s) ds \right) \frac{\eta_1}{2} \left( \lambda_\sigma \int_\theta^{1-\theta} H_\sigma(1, s) ds \right)^{-1} = \frac{\eta_1}{2}.$$

Similarly it can also be obtained that $B_2(y, z)(\vartheta) \geq \frac{\eta_1}{2}$, for $(y, z) \in \partial C_{\eta_1}$, and we get

$$\|B(y, z)\| = \|B_1(y, z)\| + \|B_2(y, z)\| \geq \|(y, z)\|. \tag{22}$$

Also for $(y, z) \in \partial C_{\eta_2}$, we get that $\|(y, z)\| = \eta_2$ for $\vartheta \in [0, 1]$. Then from $(D_2)$, one can get

$$B_1(y, z)(\vartheta) = \int_0^1 H_\sigma(\vartheta, s) \varphi(s, y(s), z(s)) ds \leq \int_0^1 H_\sigma(1, s) \varphi(s, y(s), z(s)) ds$$

$$\leq \frac{\eta_2}{2\alpha_\sigma} \int_0^1 H_\sigma(1, s) ds = \frac{\eta_2}{2}.$$

Similarly, it can also obtained that $B_2(y, z)(\vartheta) \leq \frac{\eta_2}{2}$, $(y, z) \in \partial C_{\eta_2}$. Hence, we have

$$\|B(y, z)\| = \|B_1(y, z)\| + \|B_2(y, z)\| \leq \|(y, z)\|. \tag{23}$$

Now according to the application of Lemma 6 to (20) and (21) or (22) and (23) implies that B has a fixed point $(y_1, z_1) \in \overline{C}_{\kappa, \eta}$ or $(y_1, z_1) \in \bar{C}_{\kappa_i, \eta_i} (i = 1, 2)$ such that $y_1(\vartheta) \geq \lambda_\sigma \|y_1\| > 0$ and $z_1(\vartheta) \geq \lambda_\alpha \|z_1\| > 0, \vartheta \in [0, 1]$. From which it follows that the coupled system (1) has at least one positive solution. $\square$

**Theorem 5.** *Under the conditions $(C_1) - (C_3)$ and if the following assumptions hold*

$(D_3)$ *If $\varphi^0 \alpha_\sigma < 1$; $\varphi^\infty \left( \lambda_\sigma^2 \int_\theta^{1-\theta} H_\sigma(1, s) ds \right) > 1$ and $\chi^0 \alpha_\alpha < 1$; $\chi^\infty \left( \lambda_\alpha^2 \int_\theta^{1-\theta} H_\sigma(1, s) ds \right) > 1,$*

*then the coupled system (1) has at least one positive solution. Further, if $\varphi^0 = \chi^0 = 0$ and $\varphi^\infty = \chi^\infty = \infty$, where $\lambda, H_\sigma(1, s), H_\alpha(1, s)$ defined in Lemma 8 and $\varphi_0, \chi_0, \varphi^\infty, \chi^\infty, \alpha_\sigma, \sigma_\alpha$ defined in Equations (18) and (19), then the the considered system (1) has at least one positive solution.*

**Proof.** Proof can be obtained as proof of Theorem 4. $\square$

## 4. Existence of More Than One Solutions

**Theorem 6.** *Consider that $(C_1) - (C_3)$ hold and the following conditions are satisfied:*

$(D_4)$ *If $\varphi_0 \left( \lambda_\sigma^2 \int_\theta^{1-\theta} H_\sigma(1, s) ds \right) > 1, \varphi_\infty \left( \lambda_\sigma^2 \int_\theta^{1-\theta} H_\sigma(1, s) ds \right) > 1$ and*

$$\chi_0 \left( \lambda_\alpha^2 \int_\theta^{1-\theta} H_\alpha(1, s) ds \right) > 1, \chi_\infty \left( \lambda_\alpha^2 \int_\theta^{1-\theta} H_\sigma(1, s) ds \right) > 1.$$

*Moreover, $\varphi_0 = \chi_0 = \varphi_\infty = \chi_\infty = \infty$ also hold:*

$(D_5)$ *there exists $a > 0$ such that*

$$\max_{\vartheta \in [0,1], (y,z) \in \partial C_a} \varphi(\vartheta, y, z) < \frac{a}{2\alpha_\sigma} \text{ and } \max_{\vartheta \in [0,1], (y,z) \in \partial C_a} \chi(\vartheta, y, z) < \frac{a}{2\alpha_\alpha}.$$

*Then the coupled system* (1) *has at least two positive solutions* $(y, z), (\bar{y}, \bar{z})$ *such that*

$$0 < \|(y, z)\| < a < \|(\bar{y}, \bar{z})\|. \tag{24}$$

*Where* $\lambda, H_\sigma(1, s), H_\text{æ}(1, s)$ *are defined in Lemma* 8 *and* $\varphi_0, \chi_0, \varphi^\infty, \chi^\infty, \alpha_\sigma, \sigma_\alpha$ *defined in Equations* (18) *and* (19)

**Proof.** Let $(D_4)$ hold. Select $\kappa, \eta$ such that $0 < \kappa < \mu < \eta$. Now if $\varphi_0 \left( \lambda_\sigma^2 \int\limits_\theta^{1-\theta} H_\sigma(1, s) ds \right) > 1$ and $\chi_0 \left( \lambda_\text{æ}^2 \int\limits_\theta^{1-\theta} H_\text{æ}(1, s) ds \right) > 1$, then like the proof of Theorem 4, we have

$$\|B(y, z)\| \geq |(y, z)\|, \text{ for } (y, z) \in \partial C_\kappa. \tag{25}$$

Now, if $\varphi_\infty \left( \lambda_\sigma^2 \int\limits_\theta^{1-\theta} H_\sigma(1, s) ds \right) > 1$ and $\chi_\infty \left( \lambda_\text{æ}^2 \int\limits_\theta^{1-\theta} H_\text{æ}(1, s) ds \right) > 1$, then like the proof of Theorem 4, we have

$$\|B(y, z)\| \geq \|(y, z)\|, \text{ for } (y, z) \in \partial C_\eta. \tag{26}$$

Also from $(D_5)$, $(y, z) \in \partial C_\mu$, we get

$$B_1(y, z)(\vartheta) = \int_0^1 H_\sigma(\vartheta, s) \varphi(s, y(s), z(s)) ds$$
$$\leq \int_0^1 H_\sigma(1, s) \varphi(s, y(s), z(s)) ds < \frac{\mu}{2\alpha_\sigma} \int_0^1 H_\sigma(1, s) ds = \frac{\mu}{2}.$$

Similarly, we have $B_1(y, z)(\vartheta) < \frac{\mu}{2}$ as $(y, z) \in \partial C_\mu$. Hence, we have

$$\|B(y, z)\| < |(y, z)\|, \text{ for } (y, z) \in \partial C_\mu. \tag{27}$$

Now according to Lemma 6 for (25) and (27), we have gives that B has a fixed point $(y, z) \in \partial \overline{C}_{\kappa, \mu}$ and a fixed point in $(\bar{y}, \bar{z}) \in \partial \overline{C}_{\mu, \eta}$. Therefore system (1) has at least two positive solutions $(y, z), (\bar{y}, \bar{z})$ such that $\|(y, z)\| \neq \mu$ and $\|(\bar{y}, \bar{z})\| \neq \mu$. Thus the relation (24) holds. $\square$

**Theorem 7.** *Consider that* $(C_1) - (C_3)$ *hold together with the given conditions*

$(D_6)$ $\alpha_\sigma \varphi_0 < 1$ *and* $\varphi_\infty \alpha_\sigma < 1$; $\alpha_\text{æ} \chi_0 < 1$, *and* $\chi_\infty \alpha_\text{æ} < 1$
$(D_7)$ *there exist* $\mu > 0$ *such that*

$$\max_{\vartheta \in K, (y,z) \in \partial C_\mu} \varphi(\vartheta, y, z) > \frac{\mu}{2} \left( \lambda_\sigma^2 \int\limits_\theta^{1-\theta} H_\sigma(1, s) ds \right)^{-1},$$

$$\max_{\vartheta \in K, (y,z) \in \partial C_\mu} \chi(\vartheta, y, z) > \frac{\mu}{2} \left( \lambda_\text{æ}^2 \int\limits_\theta^{1-\theta} H_\text{æ}(1, s) ds \right)^{-1},$$

*such that*

$$0 < \|(y, z)\| < \mu < \|(\bar{y}, \bar{z})\|,$$

*where* $\lambda, H_\sigma(1, s), H_\text{æ}(1, s)$ *defined in Lemma* 8 *and* $\varphi_0, \chi_0, \varphi^\infty, \chi^\infty, \alpha_\sigma, \sigma_\alpha$ *defined in Equations* (18) *and* (19). *Thus the system* (1) *has at least two positive solutions.*

**Proof.** We left the proof out, as it similar to the proof of Theorem 6. $\square$

In same line for multiple solutions we give the following results.

**Theorem 8.** *Let* $(C_1) - (C_3)$ *hold. If there exist* $2m$ *positive numbers* $\mathbf{u}_L, \hat{\mathbf{u}}_L$, $L = 1, 2 \ldots m$ *with* $\mathbf{u}_1 < \lambda_\sigma \hat{\mathbf{u}}_1 < \hat{\mathbf{u}}_1 < \mathbf{u}_2 < \lambda_\sigma \hat{\mathbf{u}}_2 < \hat{\mathbf{u}}_2 \ldots \mathbf{u}_m < \lambda_\sigma \hat{\mathbf{u}}_m < \hat{\mathbf{u}}_m$ *and* $\mathbf{u}_1 < \lambda_\text{æ} \hat{\mathbf{u}}_1 < \hat{\mathbf{u}}_1 < \mathbf{u}_2 < \lambda_\text{æ} \hat{\mathbf{u}}_2 < \hat{\mathbf{u}}_2 \ldots \mathbf{u}_m < \lambda_\text{æ} \hat{\mathbf{u}}_m < \hat{\mathbf{u}}_m$, *such that*

$(D_8)$ $\varphi(\vartheta, \mathrm{y}(\vartheta), z(\vartheta)) \geq \mathbf{u}_L \left( \lambda_\sigma \int_0^1 \mathrm{H}_\sigma(1, \mathrm{s}) \mathrm{ds} \right)^{-1}$, *for* $(\vartheta, \mathrm{y}, z) \in [0, 1] \times [\lambda_\sigma \mathbf{u}_L, \mathbf{u}_L] \times [\lambda_\text{æ} \mathbf{u}_L, \mathbf{u}_L]$, *and*

$$\varphi(\vartheta, \mathrm{y}(\vartheta), z(\vartheta)) \leq \alpha_\sigma^{-1} \hat{\mathbf{u}}_L, \text{for } (\vartheta, \mathrm{y}, z) \in [0, 1] \times [\lambda_\sigma \hat{\mathbf{u}}_L, \hat{\mathbf{u}}_L] \times [\lambda_\text{æ} \mathbf{u}_L, \mathbf{u}_L], L = 1, 2 \ldots m,$$

$(D_9)$ $\chi(\vartheta, \mathrm{y}(\vartheta), z(\vartheta)) \geq \mathbf{u}_L \left( \lambda_\text{æ} \int_0^1 \mathrm{H}_\text{æ}(1, \mathrm{s}) \mathrm{ds} \right)^{-1}$, *for* $(\vartheta, \mathrm{y}, z) \in [0, 1] \times [\lambda_\text{æ} \mathbf{u}_L, \mathbf{u}_L] \times [\lambda_\sigma \mathbf{u}_L, \mathbf{u}_L]$, *and*

$$\chi(\vartheta, \mathrm{y}(\vartheta), z(\vartheta)) \leq \alpha_\text{æ}^{-1} \hat{\mathbf{u}}_L, \text{for } (\vartheta, \mathrm{y}, z) \in [0, 1] \times [\lambda_\sigma \mathbf{u}_L, \mathbf{u}_L] \times [\lambda_\text{æ} \hat{\mathbf{u}}_L, \hat{\mathbf{u}}_L], \ L = 1, 2 \ldots m.$$

*where* $\lambda, \mathrm{H}_\sigma(1, \mathrm{s}), \mathrm{H}_\text{æ}(1, \mathrm{s})$ *defined in Lemma* 8.

*Then the coupled system* (1) *has at least* m*-positive solutions* $(\mathrm{y}_L, z_L)$, *satisfying*

$$\mathbf{u}_L \leq \|(\mathrm{y}_L, z_L)\| \leq \hat{\mathbf{u}}_L, \ L = 1, 2 \ldots m.$$

**Theorem 9.** *Suppose that* $(C_1) - (C_3)$ *holds. If there exist* $2m$ *positive numbers* $\mathbf{u}_L, \hat{\mathbf{u}}_L$, $L = 1, 2 \ldots m$, *with* $\mathbf{u}_1 < \hat{\mathbf{u}}_1 < \mathbf{u}_2 < \hat{\mathbf{u}}_2 \ldots < \mathbf{u}_m < \hat{\mathbf{u}}_m$ *such that*

$(D_{10})$ $\varphi$ *and* $\chi$ *are non-decreasing on* $[0, \hat{\mathbf{u}}_m] \ \forall \ \vartheta \in [0, 1]$;

$(D_{11})$ $\varphi(\vartheta, \mathrm{y}(\vartheta), z(\vartheta)) \geq \mathbf{u}_L \left( \lambda_\sigma \int_\theta^{1-\theta} \mathrm{H}_\sigma(1, \mathrm{s}) \mathrm{ds} \right)^{-1}$, $\varphi(\vartheta, \mathrm{y}(\vartheta), z(\vartheta)) \leq \dfrac{\hat{\mathbf{u}}_L}{\alpha_\sigma}$, $L = 1, 2 \ldots m$,

$$\chi(\vartheta, \mathrm{y}(\vartheta), z(\vartheta)) \geq \mathbf{u}_L \left( \lambda_\text{æ} \int_\theta^{1-\theta} \mathrm{H}_\text{æ}(1, \mathrm{s}) \mathrm{ds} \right)^{-1}, \ \chi(\vartheta, \mathrm{y}(\vartheta), z(\vartheta)) \leq \dfrac{\hat{\mathbf{u}}_L}{\alpha_\text{æ}}, \ L = 1, 2 \ldots m.$$

*Hence we conclude that there exist at least* m *positive solutions* $(\mathrm{y}_L, z_L)$, *corresponding to coupled system* (1) *which satisfy*

$$\mathbf{u}_L \leq \|(\mathrm{y}_L, z_L)\| \leq \hat{\mathbf{u}}_L, \ L = 1, 2 \ldots m.$$

## 5. Hyers-Ulam Stability

**Definition 6.** *[30] Let* $B_1, B_2 : E \times E \to E \times E$ *be the two operators. Then the system of operator equations*

$$\begin{cases} \mathrm{y}(\vartheta) = B_1(\mathrm{y}, z)(\vartheta) \\ z(\vartheta) = B_2(\mathrm{y}, z)(\vartheta) \end{cases} \tag{28}$$

*is called the HU stability if we can find* $J_i(i = 1, 2, 3, 4) > 0$, *with* $\text{æ}_i(i = 1, 2) > 0$ *and for each solution* $(\mathrm{y}^*, z^*) \in E \times E$ *of the inequalities given by*

$$\begin{cases} \|\mathrm{y}^* - \varphi(\mathrm{y}^*, z^*)\|_{E \times E} \leq \text{æ}_1, \\ \|z^* - \chi(\mathrm{y}^*, z^*)\|_{E \times E} \leq \text{æ}_2, \end{cases} \tag{29}$$

*there exists a solution* $(\bar{\mathrm{y}}, \bar{z}) \in E \times E$ *of system* (28) *such that*

$$\begin{cases} \|\mathrm{y}^* - \bar{\mathrm{y}}\|_{E \times E} \leq k_1 \text{æ}_1 + k_2 \text{æ}_2, \\ \|z^* - \bar{z}\|_{E \times E} \leq k_3 \text{æ}_1 + k_4 \text{æ}_2, \end{cases} \tag{30}$$

**Theorem 10.** *[30] Let* $B_1, B_2 : E \times E \to E \times E$ *be the two operators such that*

$$
\begin{cases}
||B_1(y, z) - B_1(y^*, z^*)||_{E \times E} \le k_1 ||y - y^*||_{E \times E} ds + k_2 ||z - z^*||_{E \times E} ds, \\
||B_2(y, z) - B_2(y^*, z^*)||_{E \times E} \le k_3 ||y - y^*||_{E \times E} ds + k_4 ||z - z^*||_{E \times E} ds, \\
for\ all \quad (y, z), (y^*, z^*) \in E \times E,
\end{cases}
\tag{31}
$$

*and if the matrix*

$$
\mathbf{B} = \begin{bmatrix} k_1 & k_2 \\ k_3 & k_3 \end{bmatrix}
$$

*converges to zero, then the fixed points corresponding to operator system* (28) *are HU-stable. Further, the given condition holds* $(M_{11})$ *under the continuity of* $\varphi_i, i = 1, 2$, *there exist* $f_i, H_i \in C(0, 1), i = 1, 2$ *and* $(y, z), (\overline{y}, \overline{z})$ *such that*

$$
|\varphi_i(\vartheta, y, z) - \varphi_i(\vartheta, \overline{y}, \overline{z})| \le f_i(\vartheta)|y - \overline{y}| + H_i(\vartheta)|z - \overline{z}|, i = 1, 2.
$$

*In this section, we study HU stability for the solutions of our proposed system.*

**Theorem 11.** *Suppose that the assumption* $(M_{11})$ *along with condition that matrix*

$$
\mathbf{B} = \begin{bmatrix} \displaystyle\int_0^1 H_\sigma(1, s) u_1(s) ds & \displaystyle\int_0^1 H_\sigma(1, s) v_1(s) ds \\ \displaystyle\int_0^1 H_{\text{æ}}(1, s) u_2(s) ds & \displaystyle\int_0^1 H_{\text{æ}}(1, s) v_2(s) ds \end{bmatrix}.
$$

*is converging to zero. Then, the solutions of* (1) *are HU-stable.*

**Proof.** Thanks to Theorem 2, we have

$$
\begin{cases}
||B_1(y, z) - B_1(y^*, z^*)||_{E \times E} \le \displaystyle\int_0^1 H_\sigma(1, s) u_1(s) ||y - y^*||_{E \times E} ds + \int_0^1 H_\sigma(1, s) v_1(s) ||z - z^*||_{E \times E} ds, \\
||B_2(y, z) - B_2(y^*, z^*)||_{E \times E} \le \displaystyle\int_0^1 H_{\text{æ}}(1, s) u_2(s) ||y - y^*||_{E \times E} ds + \int_0^1 H_{\text{æ}}(1, s) v_2(s) ||z - z^*||_{E \times E} ds.
\end{cases}
$$

From which we get

$$
\begin{cases}
||B_1(y, z) - B_1(y^*, z^*)||_{E \times E} \le \left[\displaystyle\int_0^1 H_\sigma(1, s) u_1(s) ds\right] ||y - y^*||_{E \times E} + \left[\displaystyle\int_0^1 H_\sigma(1, s) v_1(s) ds\right] ||z - z^*||_{E \times E}, \\
||B_2(y, z) - B_2(y^*, z^*)||_{E \times E} \le \left[\displaystyle\int_0^1 H_{\text{æ}}(1, s) u_2(s) ds\right] ||y - y^*||_{E \times E} + \left[\displaystyle\int_0^1 H_{\text{æ}}(1, s) v_2(s) ds\right] ||z - z^*||_{E \times E}.
\end{cases}
\tag{32}
$$

Analogously one has

$$
||P(y, z) - P(y^*, z^*)||_{E \times E} \le \mathbf{B} ||(y, z) - (y^*, z^*)||_{E \times E},
\tag{33}
$$

such that

$$
\mathbf{B} = \begin{bmatrix} \displaystyle\int_0^1 H_\sigma(1, s) u_1(s) ds & \displaystyle\int_0^1 H_\sigma(1, s) v_1(s) ds \\ \displaystyle\int_0^1 H_{\text{æ}}(1, s) u_2(s) ds & \displaystyle\int_0^1 H_{\text{æ}}(1, s) v_2(s) ds \end{bmatrix}.
$$

Hence, we get the required results.　□

## 6. Example

To verify the aforesaid established analysis we provide some test problems here in the given sequel.

**Example 1.** *Take the system of given BVPs with IBCs as*

$$\begin{cases} D_{+0}^{\frac{7}{2}}y(\vartheta) + \dfrac{\vartheta + 1}{4}[\Gamma(\dfrac{5}{2})|y(\vartheta)| + \cos|z(\vartheta)|] = 0, \ \vartheta \in [0,1], y, z \geq 0 \\[3mm] D_{+0}^{\frac{7}{2}}z(\vartheta) + \dfrac{\vartheta^2 + 1}{4}[\sin|y(\vartheta)| + |z(\vartheta)|] = 0, \ \vartheta \in [0,1], y, z \geq 0 \\[3mm] y(0) = y'(0) = y''(0) = 0 \quad y(1) = \dfrac{1}{2}\displaystyle\int_0^1 y(s)ds, \\[3mm] z(0) = z'(0) = z''(0) = 0 \quad z(1) = \dfrac{1}{3}\displaystyle\int_0^1 z(s)ds. \end{cases} \quad (34)$$

*Since* $\varphi(\vartheta, y(\vartheta), z(\vartheta)) = \dfrac{\vartheta + 1}{4}[\Gamma(\frac{5}{2})|y(\vartheta)| + \cos|z(\vartheta)|]$, $\chi(\vartheta, y(\vartheta), z(\vartheta)) = \dfrac{\vartheta^2 + 1}{4}[\sin|y(\vartheta)| + |z(\vartheta)|]$.

*Also as* $m = [3.5] + 1 = 4, \delta = \frac{1}{2}$ *and* $\varrho = \frac{1}{3}$.

*Then*

$$|\varphi(\vartheta, y_2, z_2) - \varphi(\vartheta, y_1, z_1)| \leq \Gamma(\frac{5}{2})\frac{\vartheta + 1}{4}|y_2 - y_1| + \frac{\vartheta + 1}{4}|z_2 - z_1|,$$

$$|\chi(\vartheta, y_2, z_2) - \chi(\vartheta, y_1, z_1)| \leq \frac{\vartheta^2 + 1}{4}|y_2 - y_1| + \frac{\vartheta^2 + 1}{4}|z_2 - z_1|.$$

*where* $u_1(\vartheta) = \frac{\vartheta + 1}{4}\Gamma(\frac{5}{2}), v_1(\vartheta) = \frac{\vartheta + 1}{4}, u_2(\vartheta) = v_2(\vartheta) = \dfrac{\vartheta^2 + 1}{4}$, *so one can get*

$$\mathbf{B} = \begin{bmatrix} \displaystyle\int_0^1 H_\sigma(1,s)u_1(s)ds & \displaystyle\int_0^1 H_\sigma(1,s)v_1(s)ds \\[4mm] \displaystyle\int_0^1 H_æ(1,s)u_2(s)ds & \displaystyle\int_0^1 H_æ(1,s)v_2(s)ds \end{bmatrix} = \begin{bmatrix} \dfrac{8}{11} & \dfrac{8}{165\sqrt{\pi}} \\[4mm] \dfrac{496}{11583\sqrt{\pi}} & \dfrac{496}{11583\sqrt{\pi}} \end{bmatrix}.$$

$$det(\mathbf{B} - \widehat{\lambda}I) = \begin{bmatrix} \dfrac{8}{11} - \widehat{\lambda} & \dfrac{8}{165\sqrt{\pi}} \\[4mm] \dfrac{496}{11583\sqrt{\pi}} & \dfrac{496}{11583\sqrt{\pi}} - \widehat{\lambda} \end{bmatrix}.$$

*We get* $\widehat{\lambda}_1 = 0.728$ *and* $\widehat{\lambda}_2 = 0.024$ *since* $æ(\mathbf{B}) = \sup\{|\widehat{\lambda}_i|, i = 1, 2\} = 0.728 < 1$. *Therefore due to Theorem 2, BVPs (34) has a unique positive solution given by*

$$\begin{cases} y(\vartheta) = \displaystyle\int_0^1 H_{\frac{7}{2}}(\vartheta, s)\frac{s + 1}{4}[\Gamma(\frac{5}{2})|y(s)| + \cos|z(s)|]ds, \\[4mm] z(\vartheta) = \displaystyle\int_0^1 H_{\frac{7}{2}}(\vartheta, s)\frac{s^2 + 1}{4}[\sin|y(s)| + |z(s)|]ds, \end{cases} \quad (35)$$

*where* $H_{\frac{7}{2}}(\vartheta, s)$ *and* $H_{\frac{7}{2}}(\vartheta, s)$ *are the Green's functions given by*

$$H_{\frac{7}{2}}(\vartheta, s) = \begin{cases} \dfrac{\vartheta^{\frac{5}{2}}(1-s)^{\frac{5}{2}}(3 + \frac{1}{2}s) - 3(\vartheta - s)^{\frac{5}{2}}}{3\Gamma(\frac{7}{2})}, & 0 \leq s \leq \vartheta \leq 1, \\[4mm] \dfrac{\vartheta^{\frac{5}{2}}(1-s)^{\frac{5}{2}}(3 + \frac{1}{2}s)}{3\Gamma(\frac{7}{2})}, & 0 \leq \vartheta \leq s \leq 1. \end{cases}$$

$$H_{\frac{7}{2}}(\vartheta,s) = \begin{cases} \dfrac{\vartheta^{\frac{5}{2}}(1-s)^{\frac{5}{2}}(\frac{19}{6}+\frac{1}{3}s) - \frac{19}{6}(\vartheta-s)^{\frac{5}{2}}}{\frac{19}{6}\Gamma(\frac{7}{2})}, & 0 \leq s \leq \vartheta \leq 1, \\[4mm] \dfrac{\vartheta^{\frac{5}{2}}(1-s)^{\frac{5}{2}}(\frac{19}{6}+\frac{1}{3}s)}{\frac{19}{6}\Gamma(\frac{7}{2})}, & 0 \leq \vartheta \leq s \leq 1. \end{cases}$$

*Further, by the use of Theorem 11, the solution is HU-stable.*

**Example 2.** *Taking a system of FODEs with IBCs as*

$$\begin{cases} D_{+0}^{\frac{10}{3}}y(\vartheta) + a(\vartheta)\sqrt{y(\vartheta)+z(\vartheta)} = 0, \ D_{+0}^{\frac{7}{2}}z(\vartheta) + b(\vartheta)\sqrt[3]{y(\vartheta)+z(\vartheta)} = 0, \ \vartheta \in (0,1), \\ y(0) = y'(0) = y''(0) = y'''(0) = 0 \quad y(1) = \displaystyle\int_0^1 y(s)ds, \\ z(0) = z'(0) = z''(0) = z'''(0) = 0 \quad z(1) = \displaystyle\int_0^1 z(s)ds. \end{cases} \tag{36}$$

*From the given system one has*

$$\varphi(\vartheta, y, z) = a(\vartheta)\sqrt{y(\vartheta)+z(\vartheta)}$$

*and*

$$\chi(\vartheta, y, z) = b(\vartheta)\sqrt[3]{y(\vartheta)+z(\vartheta)}, \ m = 4, \ \delta = \varrho = 1.$$

*Also a, b : [0,1] → [0,∞) are continuous. Now $\varphi^0 = \lim\limits_{(y,z)\to 0} \dfrac{\varphi(\vartheta,y,z)}{y+z} = \infty$, similarly $\chi^0 = \infty$.*

*Obviously we compute $\varphi^\infty = 0 = \chi^\infty$. Hence due to Theorem 4, system (36) has at least one positive solution.*

**Example 3.** *Taking another test problem with IBCs as*

$$\begin{cases} D_{+0}^{\frac{9}{2}}y(\vartheta) + (1-\vartheta^2)[y(\vartheta)+z(\vartheta)]^2 = 0, \ D_{+0}^{\frac{14}{3}}z(\vartheta) + [y(\vartheta)+z(\vartheta)]^3 = 0, \ \vartheta \in (0,1), \\ y(0) = y'(0) = y''(0) = y'''(0) = y''''(0) = 0, \quad y(1) = \dfrac{3}{2}\displaystyle\int_0^1 y(s)ds, \\ z(0) = z'(0) = z''(0) = z'''(0) = z''''(0) = 0, \quad z(1) = \dfrac{3}{2}\displaystyle\int_0^1 z(s)ds. \end{cases} \tag{37}$$

*From the considered problem (37), one has $\delta = \varrho = \frac{3}{2}$, as m = 5. It is easy to see that $\varphi^0 = \chi^0 = 0$ and $\varphi^\infty = \chi^\infty = \infty$. Therefore thanks to Theorem 5, the given system (37) has a positive solution.*

**Example 4.** *Further we take another system of FODEs with IBCs as*

$$\begin{cases} D_{+0}^{\frac{11}{2}}y(\vartheta) + \dfrac{(1+\vartheta^2)[u^2(\vartheta)+z(\vartheta)]}{(4\vartheta^2+4)\alpha_\sigma} = 0, \ \vartheta \in (0,1), \\ D_{+0}^{\frac{16}{3}}z(\vartheta) + \dfrac{(\vartheta^3+1)[y(\vartheta)+v^2(\vartheta)]}{(4\vartheta^3+4)\alpha_\ae} = 0, \ \vartheta \in (0,1), \\ y(0) = y'(0) = y''(0) = y'''(0) = y''''(0) = 0, \quad y(1) = \dfrac{3}{2}\displaystyle\int_0^1 y(s)ds, \\ z(0) = z'(0) = z''(0) = z'''(0) = z''''(0) = 0, \quad z(1) = \dfrac{3}{2}\displaystyle\int_0^1 z(s)ds. \end{cases} \tag{38}$$

*where $\delta = \varrho = \frac{3}{2}$ and m = 6. It is easy to obtain $\varphi_0 = \chi_0 = \infty$ and $\varphi_\infty = \chi_\infty = \infty$.*

*Further $\forall \, (\vartheta, y, z) \in [0,1] \times [0,1] \times [0,1]$, we have*

$$\varphi(\vartheta, y, z) \le \frac{(\vartheta^2 + 1)2}{4(\vartheta^2 + 1)\alpha_\sigma} = \frac{\alpha_\sigma^{-1}}{2}, \quad \chi(\vartheta, y, z) \le \frac{(\vartheta^3 + 1)2}{4(\vartheta^3 + 1)\alpha_\text{æ}} = \frac{\alpha_\text{æ}^{-1}}{2}.$$

*Hence all the conditions of Theorem 6 hold. Thanks to Theorem 6, the given system* (38) *has at least two positive solutions* $(y_1, z_1)$ *and* $(y_2, z_2)$ *which satisfy*

$$0 < \|(y_1, z_1)\| < 1 < \|(y_2, z_2)\|.$$

## 7. Non-Existence of Positive Solution

Here some conditions are developed under which the coupled system (1) with given IBCs has no solution.

**Theorem 12.** *Consider that* $(C_1) - (C_3)$ *hold and* $\varphi(\vartheta, y, z) < \frac{\|(y,z)\|}{2\alpha_\sigma}$ *and* $\chi(\vartheta, y, z) < \frac{\|(y,z)\|}{2\alpha_\text{æ}}$ *for all* $\vartheta \in [0,1]$, $y > 0$, $z > 0$, *then there is no positive solution for BVPs* (1).

**Proof.** Consider $(y, z)$ to be the positive solution of BVPs (1). Then, $(y, z) \in C$ for $0 < \vartheta < 1$ and

$$
\begin{aligned}
\|(y, z)\| &= \|y\| + \|z\| \\
&= \max_{\vartheta \in [0,1]} |y(\vartheta)| + \max_{\vartheta \in [0,1]} |z(\vartheta)| \\
&\le \max_{\vartheta \in [0,1]} \int_0^1 H_\sigma(\vartheta, s) |\varphi(s, y(s), z(s))| ds + \max_{\vartheta \in [0,1]} \int_0^1 H_\text{æ}(\vartheta, s) |\chi(s, y(s), z(s))| ds \\
&< \int_0^1 H_\sigma(1, s) \frac{\|(y, z)\|}{2\alpha_\text{æ}} ds + \int_0^1 H_\text{æ}(1, s) \frac{\|(y, z)\|}{2\alpha_\text{æ}} ds
\end{aligned}
$$

$$\Rightarrow \|(y, z)\| < \|(y, z)\|,$$

which is contradiction. Hence the considered system (1) has no solution. $\square$

**Theorem 13.** *Let the hypothesis* $(C_1) - (C_3)$ *hold along with the conditions*

$$\varphi(\vartheta, y(\vartheta), z(\vartheta)) \; > \; \frac{\|(y, z)\|}{2} \left( \lambda_\alpha^2 \int_\theta^{1-\theta} H_\sigma(1, s) ds \right)^{-1},$$

$$\chi(\vartheta, y(\vartheta), z(\vartheta)) \; > \; \frac{\|(y, z)\|}{2} \left( \lambda_\beta^2 \int_\theta^{1-\theta} H_\text{æ}(1, s) ds \right)^{-1}, \quad \text{for all } \vartheta \in [0,1], \; y > 0 \text{and } z > 0.$$

*Then there does not exist positive solution to BVPs* (1).

To demonstrate the results of Theorems 12 and 13 respectively, we give the following example.

**Example 5.** *Taking the given system of FODEs with given IBCs as*

$$\begin{cases} D_{+0}^{\frac{5}{2}} y(\vartheta) = 5 - 4\left(y + z + \frac{\pi}{3}\right)^{\frac{-5}{2}}, \ \vartheta \in [0,1], \\[2mm] D_{+0}^{\frac{5}{2}} z(\vartheta) = \left(30 + \frac{30}{\sqrt{y+z}}\right)^{\frac{-3}{2}} + \frac{1}{50}, \ \vartheta \in [0,1], \\[2mm] y(0) = y'(0) = y''(0) = 0, \ y(1) = \frac{1}{2}\int_0^1 y(\vartheta)d\vartheta, \\[2mm] z(0) = z'(0) = z''(0) = 0, \ z(1) = \frac{1}{2}\int_0^1 z(\vartheta)d\vartheta. \end{cases} \tag{39}$$

*Also as* $(C_1) - (C_3)$ *hold, where* $m = [2.5] + 1 = 3$ *and* $\delta = \varrho = \frac{1}{2}$. *We calculate*

$$\varphi^0 = 5 - \left(\frac{3}{\pi}\right)^{\frac{5}{2}}, \ \chi^0 \ = \ \frac{1}{50}, \ \varphi^\infty = 5, \ \chi^\infty = 51,$$

$$\left(5 - \left(\frac{3}{\pi}\right)^{\frac{5}{2}}\right)\|(y,z)\| \ < \ \varphi(\vartheta, y(\vartheta), z(\vartheta)) < 5\|(y,z)\|,$$

$$\frac{1}{50}\|(y,z)\| \ < \ \chi(\vartheta, y(\vartheta), z(\vartheta)) < 51\|(y,z)\|.$$

*Therefore we have*

$$5 - \left(\frac{3}{\pi}\right)^{\frac{5}{2}}\|(y,z)\| < \varphi(\vartheta, y(\vartheta), z(\vartheta)) < 5\|(y,z)\| \text{ and } \chi(\vartheta, y(\vartheta), z(\vartheta)) < 5\|(y,z)\| < \frac{\|(y,z)\|}{\alpha_\sigma},$$

*where* $\alpha_\sigma \approx 0.32239$ *and* $\alpha_æ \approx 0.32239$.
**Case I:** *Now*

$$\varphi(\vartheta, y(\vartheta), z(\vartheta)) < \frac{\|(y,z)\|}{\alpha_\sigma} \approx 1.1413\|(y,z)\|$$

*yields that*

$$\varphi(\vartheta, y(\vartheta), z(\vartheta)) < 5\|(y,z)\| \approx 3.1018\|(y,z)\|$$

*and*

$$\chi(\vartheta, y(\vartheta), z(\vartheta)) < 51\|(y,z)\| \approx 3.1018\|(y,z)\|.$$

*Hence under the condition of Theorem 12, there is no solution corresponding to problem* (39).
**Case II:** *Also*

$$\varphi(\vartheta, y(\vartheta), z(\vartheta)) > \left(5 - \left(\frac{3}{\pi}\right)^{\frac{5}{2}}\right)\|(y,z)\| > \|(y,z)\| \left(\lambda_\alpha^2 \int_{\frac{1}{100}}^{\frac{99}{100}} H_\alpha(1,s)ds\right)^{-1} \approx 0.615\|(y,z)\|$$

*and*

$$\chi(\vartheta, y(\vartheta), z(\vartheta)) > \frac{1}{50}\|(y,z)\| > \|(y,z)\| \left(\lambda_æ^2 \int_{\frac{1}{100}}^{\frac{99}{100}} H_æ(1,s)ds\right)^{-1} \approx 0.615\|(y,z)\|.$$

*Hence under the condition of Theorem 13, there is no solution corresponding to coupled system* (39).

## 8. Conclusions

In the above research work we have successfully investigated a coupled system of nonlinear FODEs with IBCs for multiplicity results. Further, the aforesaid investigation has been strengthened by developing some conditions under which the solutions of the proposed system are HU-stable. Further some results which demonstrate the conditions of nonexistence of solutions have been established. The whole results have been verified by considering some examples where needed.

**Author Contributions:** All authors have equal contribution in this paper.

**Funding:** This research work has been supported financially by KMUTT Fixed Point Research Laboratory, KMUTT-Fixed Point Theory and Applications Research Group, Faculty of Science, King Mongkut's University of Technology Thonburi (KMUTT), 126 Pracha-Uthit Road, Bang Mod, Thrung Khru, Bangkok 10140, Thailand.

**Acknowledgments:** All authors are very thankful to the referees for their useful comments and suggestions.

**Conflicts of Interest:** The authors declare no conflict of interest.

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
