# Peer review of "On Ulam Stability and Multiplicity Results to a Nonlinear Coupled System with Integral Boundary Conditions"

_mathematics, doi:10.3390/math7030223_

Round 1
Reviewer 1 Report
The paper is very interesting and well written. Results are presented in a clear manner, so that each step can be easily followed. The general impression of the paper is positive, reporting new results and as well justifying them by means of suitable examples.
There are only a very few points to be clarified or fixed, mainly typos, which do not affect the general positive opinion about the paper and its publication.
- In the line just after line 29, the author "wang" should be written with a capital letter at the beginning.
- In the same line, the IBVP acronym is quite self-explaining, but not explicitly introduced before; I would suggest adding the full name before as done for the others acronyms.
- In Definition 2.1, page 3, there is a "u" missing in the spelling of "Liouville"'s name.
- In line 62, I guess the second "y" appearing should be written in italic, as in the following lines.
- In Definition 2.7, page 4, the two indexes (m,m) for M and R should both be written either in italic or in normal font.
- In line 100, "consider" should be written with a capital letter at the beginning.
- In line 134, "solutions" should be written as a singular word, not plural.
- In line 150, "is" should be removed.
- In the second line after line 175, "systems" should be written as a singular word, not plural.
- In Theorem 5.2, page 14, "Under" should be written with a small letter at the beginning rather than capital.
- In line 189, "problem" should be written as a plural word rather than singular.
- In Example 6.1, page 15, m should be [3.5] rather than [3.3].
- In Example 6.2, page 16, m should be 4 instead of 5.
- In Example 6.3, page 16, m should be 5 instead of 6.
Author Response
We have revised our paper as suggested by the referee.

Reviewer 2 Report
This has nice end results on two coupled fractional order partial differential equations under somewhat restrictive conditions. However the presentation could use some improvements for readability/accuracy.
Overall the English could use attention of a technical language editor. Beside use of "a" and "the" there are statements such as (p. 3 line 40) "... in the last we ..." probably should be "... in the end we ..."'
There are inconsistencies in use of symbols. For example the phi at the top of page 2 is the negative of the same symbol phi at equation (1).
[all displayed equations could use numbering for ease of reference]
Definitions would best come prior to their use. For example at the key equation (1) the Riemann-Liouville fractional derivative is used but that is defined on a later page [where it could use an equation number].
Some symbols are taken as known and probably standard, as Phi of line 61, but others are confusing as Mm,m of line 67 which apparently is not the m,m entry in M.
At various places "convergence" is mentioned though the conditions under which this happens and the type of convergence could use clarification .
The definitions of page 6 come rather late and seem to over-ride earlier ones, so it would be best for a reader to know from the very start, what definitions are to hold throughout.
At line 111 and elsewhere "positive" solutions are mentioned but a definition is not found and important since this is for vectors.
After (15) P is apparently previously defined but I do not find so it should say where it is defined.
Around pages 10-13, the alphas seem important so perhaps some interpretation of them and some of the other rather large number of symbols would help a reader understand all the formulas.
It is not clear as to the meaning of "two positive solutions". It appears that it may be that one of y in a pair (y,z) and z in another pair qualifies as well as one pair with both y and z positive?
At line 175 it would help to give the names that go with HU.
The examples seem to have a number of misprints so should be double checked. For example: After (28) maybe m=[3.5] instead of [3.3] and after that 1/3 and 1/2 are reversed and m=6 at (31) should be m=5.
The means to compute rho(B) at line 191 would help. And at line 198 presentation of an actual solution would greatly add to being able to interpret the theory.
Author Response
We have replied to the points raised by the referee. We are very thankful for the suggestions.
